# Inferring evolutionary pathways and directed genotype networks of foodborne pathogens

Oliver M. Cliff[1], Natalia McLean[1], Vitali Sintchenko[2,3], Kristopher M. Fair[1], Tania C. Sorrell[2,3], Stuart Kauffman[4], Mikhail Prokopenko[1,2]*

**1** Centre for Complex Systems, Faculty of Engineering, University of Sydney, Sydney, New South Wales, Australia, **2** Marie Bashir Institute for Infectious Diseases and Biosecurity, University of Sydney, Westmead, New South Wales, Australia, **3** Centre for Infectious Diseases and Microbiology – Public Health, Institute of Clinical Pathology and Medical Research, NSW Health Pathology, Westmead Hospital, Sydney, New South Wales, Australia, **4** Institute for Systems Biology, Seattle, Washington, USA

\* mikhail.prokopenko@sydney.edu.au

**Data Availability Statement:** The code and data used in this study are available at GitHub Digital Repository: https://github.com/olivercliff/inferring-networks-salmonella.

## Abstract

Modelling the emergence of foodborne pathogens is a crucial step in the prediction and prevention of disease outbreaks. Unfortunately, the mechanisms that drive the evolution of such continuously adapting pathogens remain poorly understood. Here, we combine molecular genotyping with network science and Bayesian inference to infer directed genotype networks—and trace the emergence and evolutionary paths—of *Salmonella* Typhimurium (STM) from nine years of Australian disease surveillance data. We construct networks where nodes represent STM strains and directed edges represent evolutionary steps, presenting evidence that the structural (i.e., network-based) features are relevant to understanding the functional (i.e., fitness-based) progression of co-evolving STM strains. This is argued by showing that outbreak severity, i.e., prevalence, correlates to: (i) the network path length to the most prevalent node ($r = -0.613$, $N = 690$); and (ii) the network connected-component size ($r = 0.739$). Moreover, we uncover distinct exploration-exploitation pathways in the genetic space of STM, including a strong evolutionary drive through a transition region. This is examined via the 6,897 distinct evolutionary paths in the directed network, where we observe a dominant 66% of these paths decrease in network centrality, whilst increasing in prevalence. Furthermore, 72.4% of all paths originate in the transition region, with 64% of those following the dominant direction. Further, we find that the length of an evolutionary path strongly correlates to its increase in prevalence ($r = 0.497$). Combined, these findings indicate that longer evolutionary paths result in genetically rare and virulent strains, which mostly evolve from a single transition point. Our results not only validate our widely-applicable approach for inferring directed genotype networks from data, but also provide a unique insight into the elusive functional and structural drivers of STM bacteria.

## Author summary

We study emergence and evolution of foodborne pathogens, and provide a new method for public health surveillance dealing with genetically diverse and spatiotemporally

**Funding:** This research was funded by The Australian Research Council grant DP200103005 (MP, VS, TCS and SK). VS acknowledges funding from a National Health and Medical Research Council grant (APP1123879) and TCS acknowledges a Centre of Research Excellence grant (APP1102962). TCS is a Sydney Medical Foundation Fellow. The funders had no role in study design, data collection and analysis, decision to publish, or preparation of the manuscript.

**Competing interests:** The authors have declared that no competing interests exist.

distributed epidemic scenarios. The proposed method interprets the surveillance data through genotype networks, and discovers how the most dominant strains of infection emerge and adapt. The approach allows us to correlate the strength of epidemics with genetic features of observed pathogens. This could open a way to predict and contain epidemics closer to their source, enabling more timely and precise allocations of public health resources, as well as efficient interventions during epidemics. This should make a significant economic and social impact, improving health of the population, while also safeguarding national and international supply chains.

This is a *PLOS Computational Biology* Methods paper.

## Introduction

Rapidly escalating outbreaks of infectious diseases directly and unexpectedly impact the lives of people. The consequences are quickly seen in health and behavior, labour supply and productivity, trade across regions, and in consumer and business confidence [1–3]. Foodborne infections remain one of the highest-priority targeted diseases by the World Health Organization due to the emergence of the antimicrobial resistance strains [4]. Of particular importance is the *Salmonella enterica* subspecies *enterica* servora Typhimurium (STM)—the dominant cause of salmonellosis—which is responsible for over 155,000 deaths with a estimated 93.8 million human cases of the disease each year [5–7].

Since diverging from a common ancestor shared with *E. coli*, Salmonella co-evolved with animal hosts for millions of years [8], and there are several more recent historical drivers catalyzing this process. A shift from foraging to agriculture and domestication of animals led to a notable increase in the risk of zoonotic diseases caused by interactions between wildlife and livestock [9]. Industrialization and urbanization improved nutrition and standards of living. We are now witnessing globalization during the Anthropocene transition, with a significant spread of infections including rapidly mutating and drug resistant pathogens [10–12].

The constant evolution of STM, as well as its relentless adaptation within different ecological niches, have produced its significant diversity, allowing for the proliferation of highly adapted and virulent strains [13–15]. A continuing evolution of *Salmonella* within hosts is driven by genomic variations that may occur during each infection, triggering adaptations to a new niche—this process may result from changes in agricultural practices, the use of antimicrobial agents, or presence of immunocompromised hosts [8]. It has been observed that the diversity of the STM influences the regional prevalence in individual strains [15–18], although the driving forces within the STM population remain poorly understood [18].

The increasing complexity of disease transmission pathways creates a major bio- and health security threat, impeding timely outbreak detection and an efficient containment close to the source. Understanding the underlying mechanisms of such complexity remains a substantial empirical and theoretical challenge, exacerbated by multiple scales of the evolution. While slow bacterial specialization increases genetic variance over a longer time-scale, the rapid, sudden shifts in bacterial populations exhibit genetic branching and may cause abrupt transitions. To this end, the main aim of this study is in tracing the emergence and evolutionary paths of continuously adapting foodborne pathogens.

We develop a framework for inferring directed genotype networks from disease surveillance data by combining high-resolution molecular genotyping with network science and Bayesian inference. The resultant networks are used to relate the outbreak severity (average

strain prevalence) and network attributes (e.g., closeness centrality), with certain patterns in network dynamics revealing a strong evolutionary drive through a transition region. They allowed us to quantify several key correlations with respect to the size of genetic clusters and the length of evolutionary paths. These findings not only offer important insights into the evolution of foodborne pathogens, but also provide rigorous tools for high-precision epidemic modeling and prediction.

## Emergence and evolution of dominant strains

Networks and various topological measures have been extensively used to model and quantify interactions, as well as effects of those interactions on a global scale, in a variety of biological, evolutionary and epidemiological contexts, ranging from molecules and genes, to individual organisms [19–26]. The ability to function effectively arises in complex networks not from individual nodes, but rather from the way they interact and process information [27–29]: the function is constrained by the structure, and the structure reconfigures due to function. In this study we consider the evolutionary aspect of epidemics as part of the "function" (i.e., fitness), ascribed to rapidly adapting pathogens. We then relate this function to the "structure" revealed in the network topology, formed by connecting genotypes of microorganisms observable over time.

The accumulation of high-resolution genotyping data from public health surveillance has opened new opportunities to examine emergence and evolution of dominant strains. While relationships between infection cases during an outbreak have been previously represented as pairwise distances and visualized as unweighted or weighted graphs [25, 30], the evolution of pathogens rapidly occupying a niche and transitioning to an epidemic strength has not been explained in terms of fine-grained evolutionary paths, inferred from longitudinal data. Thus, our specific aim is to identify distinct evolutionary pathways within the STM population, and show that emergence of virulent STM strains is guided by an evolutionary drive, rather than with random sets of co-circulating STM genotypes [31, 32]. In doing so, we propose a general network-based model of pathogens' emergence and evolution, where directed genotype networks are inferred from disease surveillance data.

Using the model, we demonstrate that the network properties of the STM population as a whole can be correlated with the strength of observed outbreaks. We argue that an ongoing "arms race" between pathogens and other microorganisms trying to occupy the same niche, while resisting immune defenses of the host, creates specific evolutionary drives. Our network-based model formalizes this notion, offering a generic definition of genetic distance between observed strains, coupled with a measure of their temporal synchrony. As such, it can be applied to different pathogens and infectious diseases, genetically tracked over time.

Plasticity of genomic material is a fundamental biological property of microorganisms, and so pathogen populations can be categorized into discrete genotypes according to genetic loci, with a notable effect on the transmission. Often, new epidemics are triggered by novel genotypes that evade the acquired immunity within the host population created by their predecessors, e.g., influenza [33, 34] and other coronaviruses [35, 36]. Isolates from epidemiologically linked hosts typically show indistinguishable DNA patterns, suggesting that high-resolution molecular genotyping may offer an efficient approach to tracing foodborne epidemics and identifying their origins.

In this study we focus on emergence and evolution of STM strains in Australia, where Salmonella remains the most important cause of severe foodborne disease, with considerable morbidity and economic cost: the annual costs of foodborne diseases to the national economy circa 2000 were estimated as A\$1.249 billion annually [37, 38]. Moreover, the number of

salmonellosis cases in Australia has doubled over the last decade despite increasing efforts in public health control and prevention, and the notification rate exceeded 70 per 100,000 [39, 40]. In fact, Australia has more outbreaks of Salmonella than any other country [40]. Even so, the notified and confirmed cases are only the "tip of the iceberg", with about 9% of cases notified to public health authorities [6]. This greatly increases the risk, especially for vulnerable groups, as foodborne infections are more likely to be fatal in people with compromised immune systems, babies and the elderly.

Our analysis is carried out using high-resolution molecular "fingerprint" data on STM pathogens and associated epidemics in New South Wales (NSW) since 2008, a unique set of longitudinal data covering a decade of surveillance. This dataset comprises a collection of 17,005 isolates of non-typhoidal Salmonella, identified in the NSW State Enteric Reference Laboratory in Sydney, Australia, between January 1, 2008 and December 31, 2016. Each isolate was genotyped using multiple-locus variable-number tandem-repeats analysis (MLVA). This dataset constitutes 98.7% of all isolated STM cases identified in NSW over the 3,287 day period. The prevalence of a specific MLVA profile is obtained by adding up all occurrences of the profile in the entire dataset. There are 1,673 unique MLVA profiles in the dataset considered in our study: importantly, new MLVA profiles were detected regularly (uniformly distributed) throughout the observational period, as demonstrated by Fig 1, which shows a diagonal outline over the profiles sorted according to the date of their first detection.

The ability to predict STM activity up to several hundred days in advance by monitoring changes in the diversity of the STM strains has been demonstrated in our previous work [25]. This was achieved by characterizing the outbreaks with non-linear but symmetric (undirected) interactions between the strains. In this study, we consider both the temporal and genetic proximity between strains, aiming to establish evolutionary directions of genotype changes, as well as their paths. Fig 2 illustrates the directed genotype network inferred from the STM incidence data, based on both the genetic and temporal distances between MLVA profiles.

The proposed network-centered view on the evolution of foodborne pathogens allows us to identify and quantify preferential evolutionary pathways. In doing so we define a space mapping both the relative prevalence and spread of the pathogens. Then we examine several possible directions in this space, and identify dominant trends across these directions.

## Model

We examine the emergence and evolution of strain properties using the dataset of genotyped STM strains identified by MLVA. In MLVA, the "tandem-repeat" identifies the number of repeated units for a particular nucleotide pattern in a genome. The numbers of repeats in several fixed genetic locations (loci) are combined into a sequence of integers, e.g., 3-9-7-12-523, and this sequence is referred to as the MLVA profile (see Materials and methods for details). Crucially, MLVA profiles are similar for epidemiologically linked cases but differ between bacterial strains, ensuring a sufficient discriminatory power required for public health laboratory surveillance and outbreak investigations [31, 41].

We utilize this dataset to infer a directed network where each node represents a unique strain. Directed edges, connecting "predecessor" and "successor" profiles, are created on the basis of both genetic and temporal proximity, accounting for (i) the genetic difference between the repeated units for the respective loci, and (ii) temporal difference between the date of the successor's first detection and an observed occurrence of the predecessor during a period before (or shortly after) this date. The temporal threshold of an edge requires that an existing strain, within genetic proximity, occur within a certain number of days ("pre-window") of the new strain, or no-more than a certain number of days after it ("post-window"). The post-

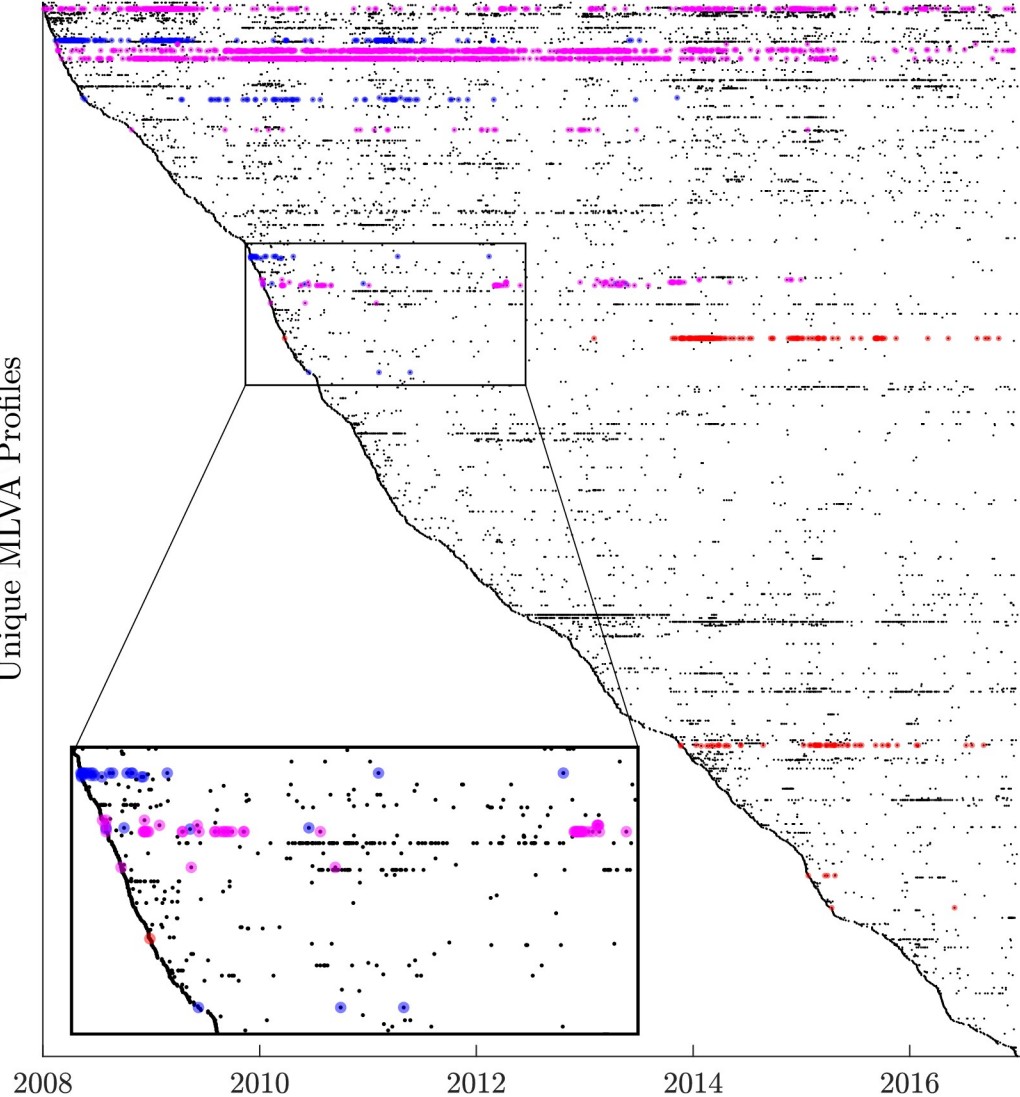

**Fig 1. The 1,673 unique MLVA profiles isolated between 1 January 2008 and 31 December 2016.** Each horizontal line shows the temporal history of a given MLVA profile, with each point showing the day a sample was collected. The dataset contains 17,005 cases of STM, which comprises 98.7% of all isolated cases in NSW over this 3,287 day period. The profiles are sorted according to the date of their first detection—resulting in a diagonal outline—indicating that new MLVA profiles are usually detected regularly throughout the observational period. Once detected, the profiles typically continue to be observed in the dataset. Three actual case studies associated with outbreaks previously investigated by the Australian Department of Health (see Materials and methods) are highlighted in different colors, illustrating distinct MLVA profiles that are grouped according to their genetic and temporal proximity (that is, profiles with the same color lie on the same inferred evolutionary path).

window accounts for possible delays in case isolation, e.g., from a delay in symptom-onset or analysis. For two strains that are genetically close and have their first instances detected within the post-window of each other this may lead to a bidirectional network edge. The Bayesian inference algorithm that dynamically identifies plausible temporal pre- and post-windows, given the genetic distance between predecessor and successor profiles, is described in Materials and methods. Thus, the directionality of the inferred edge is determined by the chronological synchrony of the reported STM cases, given their MLVA difference. By building the edge set

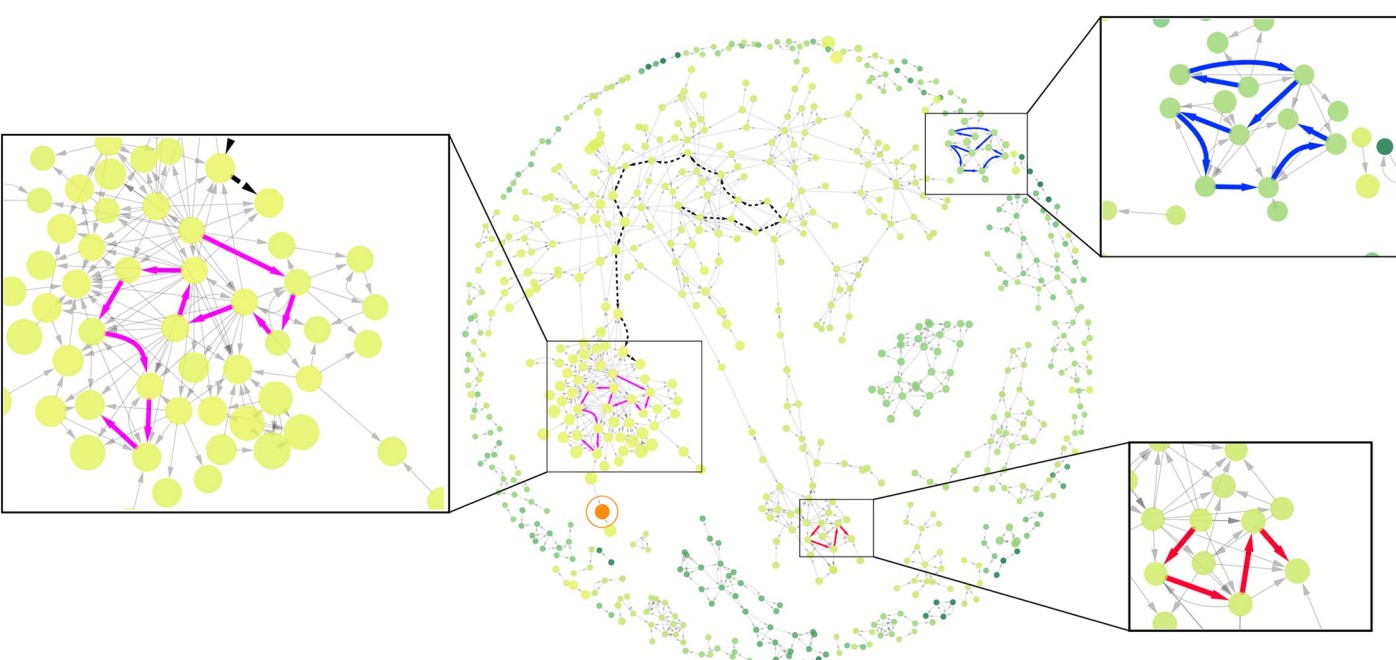

**Fig 2. The directed network based on genetic and temporal distances between MLVA profiles, where each directed edge is obtained from Bayesian inference.** The network comprises 690 connected nodes and 1,020 directed edges, with each node representing an MLVA profile and each edge representing an inferred evolutionary step. In addition, each profile may be associated with a cluster of other profiles within a genetic distance $G$. Formally, the cluster $C_i$ associated with node $i$ is given by $C_i = \{j: G_{ij} \leq G_{max}\}$, for some threshold $G_{max}$ (Materials and methods). The size of the nodes represents the average prevalence within this cluster approximating the average virulence across closely related strains. In this network, the node associated with the most prevalent cluster (profile 3-9-7-17-523) is highlighted and circled in orange. The color of all other nodes illustrates their genetic distance to this reference node, from light green (closest) to dark (farthest). The layout of nodes is given by Cytoscape's edge-weighted spring-embedded algorithm, which groups together cliques of nodes. From this graph, we can capture a potential evolutionary path between two connected nodes as a sequence of directed edges between them, as described in Model. We highlight three case studies as paths in magenta, blue, and red that contain eleven, nine, and five STM strains respectively. These paths include profiles associated with outbreaks previously investigated by the Australian Department of Health (see Materials and methods). The path shown in magenta includes (as the 9th node) profile 3-10-7-14-523, which caused a significant outbreak in October 2013. The origin node of the blue path is profile 3-12-9-10-550, which was the source of an outbreak in January 2011. The origin profile of the red path is 3-16-9-11-523, which is associated with at least two outbreaks in February 2014 and September 2015. In addition to the case studies above, we illustrate the longest path (17 nodes) inferred from the dataset as a dashed black line with a source node near the top-left of the network. The origin of this path is profile 3-12-9-12-523, which was consistently detected from January 2008 to November 2016, spanning the entire dataset.

of potential predecessors for each node in the network (profile), we obtain the directed genotype network.

The dataset exhibits a heavy-tailed distribution of STM strains, in terms of frequency vs prevalence, with most prevalent MLVA profiles occurring less frequently than other profiles [25], and so we cluster similar profiles together in order to denoise the data. For each profile, an overlapping clustering algorithm groups all neighbours of the profile within a certain threshold distance $G_{max}$, as described in Materials and methods. This allows us to compute the average cluster prevalence approximating the average virulence across genetically close strains. This clustering produces an underlying undirected network, where we characterize each node (MLVA profile) in terms of two quantities: (i) the average prevalence of the immediate cluster of the node, that is, the average prevalence of genetically close MLVA profiles, (ii) the centrality of the node within the corresponding undirected genetic network (where network centrality is suitably defined, see Materials and methods). The average prevalence of a network node (MLVA profile) represents the average strength of outbreaks associated with the genetically close strains. On the other hand, the centrality of the node within the entire weighted network quantifies how genetically connected it is overall with other strains.

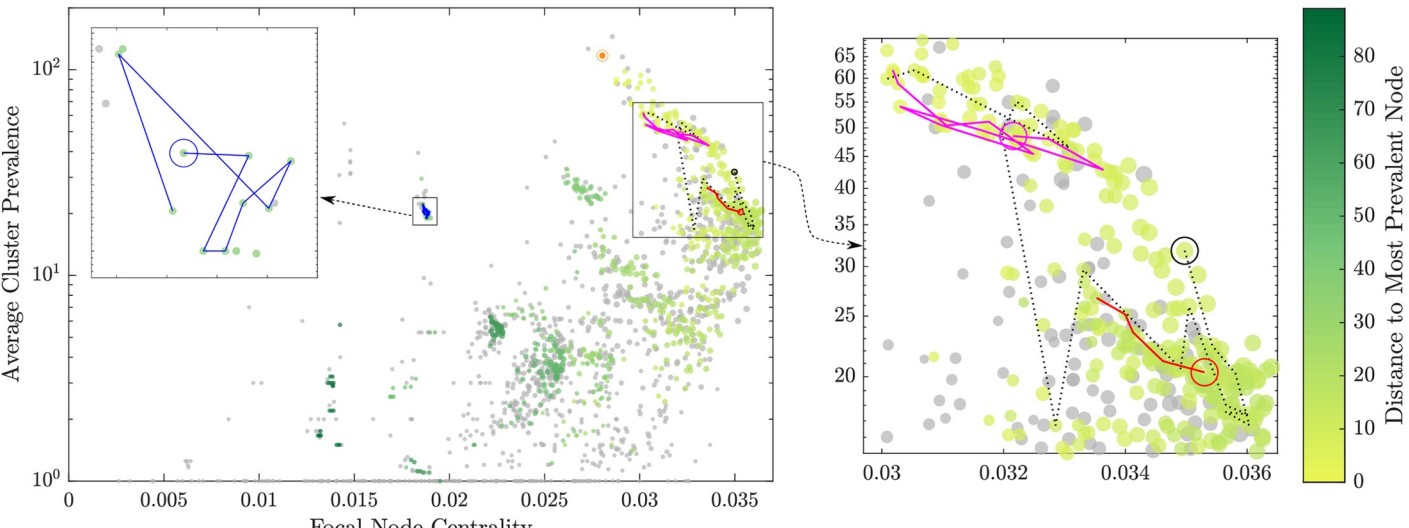

**Fig 3. Structure-function relationship in the genotype networks, mapping the centrality of profiles (x-axis) and their average cluster prevalence (y-axis).** The size of each point is proportional to the size of its corresponding cluster. The centrality is measured with respect to the complete undirected genotype network. The colour shows the genetic distance of a profile to the reference profile (in orange, see Fig 2), with grey points indicating disconnected profiles. The gradient of colouring indicates an evolutionary trend from low-to-high centrality (left to right) and then a reverse in direction from high to medium centrality (right to left). During this trend, the average prevalence of the node clusters is increasing. Consequently, two branches can be distinguished: (i) from weakly-prevalent and peripheral nodes (the bottom-left corner), towards a region comprising the profiles with high centrality (between 0.033 and 0.036) and medium cluster prevalence (between 12 and 32.1), and (ii) from this region towards the reference profile (in orange), in the direction of increasing prevalence (bottom to top) but decreasing centrality (right to left). The three case studies are shown in magenta, blue, and red, as well as the longest path (dashed black), with the source profile of these paths circled with the colour of the path.

Analysis of the undirected network produces the prevalence-centrality space, on which we can project directed evolutionary paths. In other words, once the directed genetic network is inferred through the process described above, we can identify directed paths within the network, where each path links a series of STM strains that represent a sequence of potential evolutionary steps from ancestors to descendants. If a path is sufficiently long, these genetic changes (steps) may span a fairly long temporal period, and connect distant STM MLVA profiles, separated by a number of evolutionary/adaptive steps. In order to identify distinct evolutionary paths, only unique paths are distinguished, with all sub-paths discarded.

In order to quantify the emergence and evolution along these genetic branches, we construct and trace all evolutionary paths, formed by connecting a sequence of directed edges. Formally, a path is a sequence of distinct directed edges each of which links two distinct connected nodes in the inferred directed network. This complements the quantification of centrality obtained over the undirected network, by augmenting the centrality-prevalence space with nodes which appear in the directed network. In other words, while every MLVA profile is present in the centrality-prevalence space as a node in the undirected network, only some of the strains are highlighted as nodes signifying detected genetic steps (colored nodes in Fig 3). Furthermore, only a subset of these nodes may appear in sufficiently long paths, while some may be limited to very short predecessor-successor steps.

Each path is traced in the centrality-prevalence space, by comparing its start and end points, in terms of two characteristics: (i) the relative cluster prevalence; (ii) the relative centrality, each of which may increase or decrease from start to end. An evolutionary drive is expected to increase the average cluster prevalence along a path, denoted BT (bottom-to-top), while the opposite direction (TB) would indicate a loss of fitness. We will refer to the paths developing in BT direction as *successful* evolutionary paths (since the average cluster

prevalence is increased along the path), and to the paths following TB direction as *unsuccess-ful* evolutionary paths (as the cluster prevalence is decreased). A change in centrality may also take two directions: left-to-right (LR, increasing centrality along the path), and right-to-left (RL, decreasing centrality). The successful evolutionary paths that develop in the RL direction will be considered *exploitative*, as the average cluster prevalence is increased at the expense of the corresponding decrease in centrality. On the other hand, the successful evolu-tionary paths which follow LR direction will be considered *explorative*, since an increase in the cluster prevalence is accompanied by an increase in centrality. Tracing all possible unique paths can reveal a particular signature in the centrality-prevalence space, relating the function (i.e., fitness) and structure (i.e., centrality), and characterizing dominant evolution-ary pathways in network terms.

## Results

While pursuing our main aim of characterizing the general direction of the STM evolution and its salient pathways, we illustrate the results with a study of three community outbreaks of STM. These cases relate to outbreaks previously investigated by the Australian Department of Health: (i) STM MLVA profile 3-10-7-14-523, which was first reported in January 2010 and later caused a significant outbreak in October 2013; (ii) MLVA 3-12-9-10-550, which was first reported in February 2008, and was later responsible for the outbreak in January 2011; and (iii) profile 3-16-9-11-523, which first appeared in March 2010, and was associated with at least two outbreaks in February 2014 and September 2015. In addition to these cases, we trace the longest path (with 17 nodes) inferred from the dataset, originating from MLVA profile 3-12-9-12-523, which was consistently detected from January 2008 to November 2016, i.e., this path spans the entire interval captured by the dataset.

The inferred genotype network comprises 690 connected nodes (MLVA profiles) and 1,020 directed edges (evolutionary steps), while 983 nodes remain disconnected. Fig 2 visualizes the corresponding paths within the entire network. The same paths in the centrality-prevalence space are presented in Fig 3, where the prevalence is averaged over a cluster of neighboring profiles within a genetic distance $G_{max} = 5$, while the centrality is measured with respect to the complete undirected genotype network.

In prior work, we demonstrated that the average strain prevalence is correlated with the strain centrality [25]. However, this relationship is non-linear, with the most prevalent clusters showing medium centrality, while nodes with the highest centrality exhibit medium average cluster prevalence (Fig 3). Moreover, there are two branches distinguishable in the space formed by node centrality and cluster prevalence. The first, lower, branch develops from weakly-prevalent and peripheral nodes (the bottom-left corner of Fig 3), towards a region comprising the profiles with high centrality and medium cluster prevalence, i.e., node central-ity between 0.033 and 0.036, and cluster prevalence between 12 and 32.1. This region is highlighted in Fig 4. It will be specifically analyzed later on, in terms of a transition to the sec-ond, upper, branch which continues from this region in the direction of increasing prevalence (bottom to top) and decreasing centrality (right to left), towards a reference profile shown in orange in Fig 3. We find that the gradient of coloring, set with respect to the most prevalent cluster (MLVA 3-9-7-17-523), i.e., our reference point, indicates the presence of an evolution-ary pathway along the upper branch: from the highly-central nodes with medium cluster prevalence towards the most prevalent and very "successful" clusters (i.e., right-to-left and bot-tom-to-top). In addition, the gradient highlights a somewhat punctuated evolution along the lower branch from the weakly-prevalent periphery toward highly-central, but not yet most

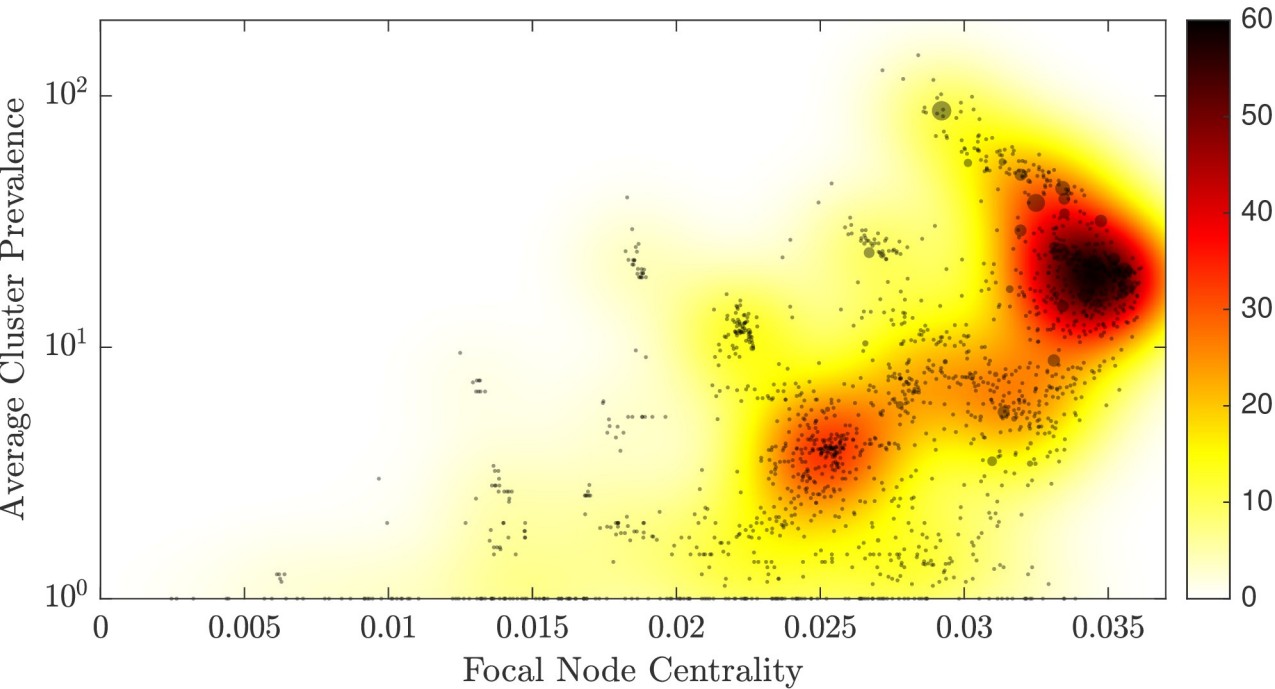

**Fig 4. The structure of the centrality-prevalence space, revealed by the expected value of the positive gain in prevalence.** For every MLVA profile, the point size is shown in proportion to the average change in prevalence. The size is set to one if the change is negative, so that the contribution of all unsuccessful paths with negative change in prevalence is bounded. Colour intensity indicates the estimated density, given the average changes in prevalence for all profiles. The transition region (intense red colour) is shaped by the profiles with high centrality and medium cluster prevalence, i.e., node centrality between 0.0329 and 0.0359, and cluster prevalence between $10^{1.14}$ and $10^{1.46}$. The evolutionary paths originating from profiles in this region tend to produce the higher average change in prevalence, that is, these paths develop from right to left and from bottom to top.

dominant, nodes (i.e., left-to-right and bottom-to-top). This branch includes several separated plateaus with mostly flat cluster prevalence.

Crucially, the centrality-prevalence space reveals that the branching occurs precisely at the highly-central nodes, suggesting that a typical evolutionary path initially increases the average strain prevalence by *exploring* the genetic space and making more genetic connections (thus, increasing the centrality of the node). Then, after reaching a central place in the genetic network, the evolution takes a turn toward *exploiting* the genetic space: at this stage, an evolving strain becomes more distant from its genetic relatives, while increasing the average prevalence of its immediate cluster. In other words, a reduction in the node centrality within the network corresponds to a better occupation of the niche (stronger adaptation), coinciding with a significant growth in the STM transmissibility and resultant size of the associated outbreaks.

In order to further verify this observation, we trace the relative change in prevalence along both successful and unsuccessful paths. Fig 4 visualises the expected value of the positive gain in cluster prevalence, while bounding the contributions of all unsuccessful paths to 1 (using kernel density estimation [42]; the accuracy of the density estimation is shown in S1 Fig). The points with higher than 75% of the maximum expected value form a well-defined region within the centrality-prevalence space. Specifically, it is shaped by high centrality (i.e., between 0.033 and 0.036) and medium cluster prevalence (i.e., between 12 and 32.1), and so emerges as the *transition* region. The evolutionary paths starting from profiles contained within this region tend to produce a higher average change in prevalence and are thus considered more successful. Importantly, as the structure on the upper genetic branch is constrained, these paths develop from right-to-left, exploiting the niche (RL-BT). The origins of explorative paths

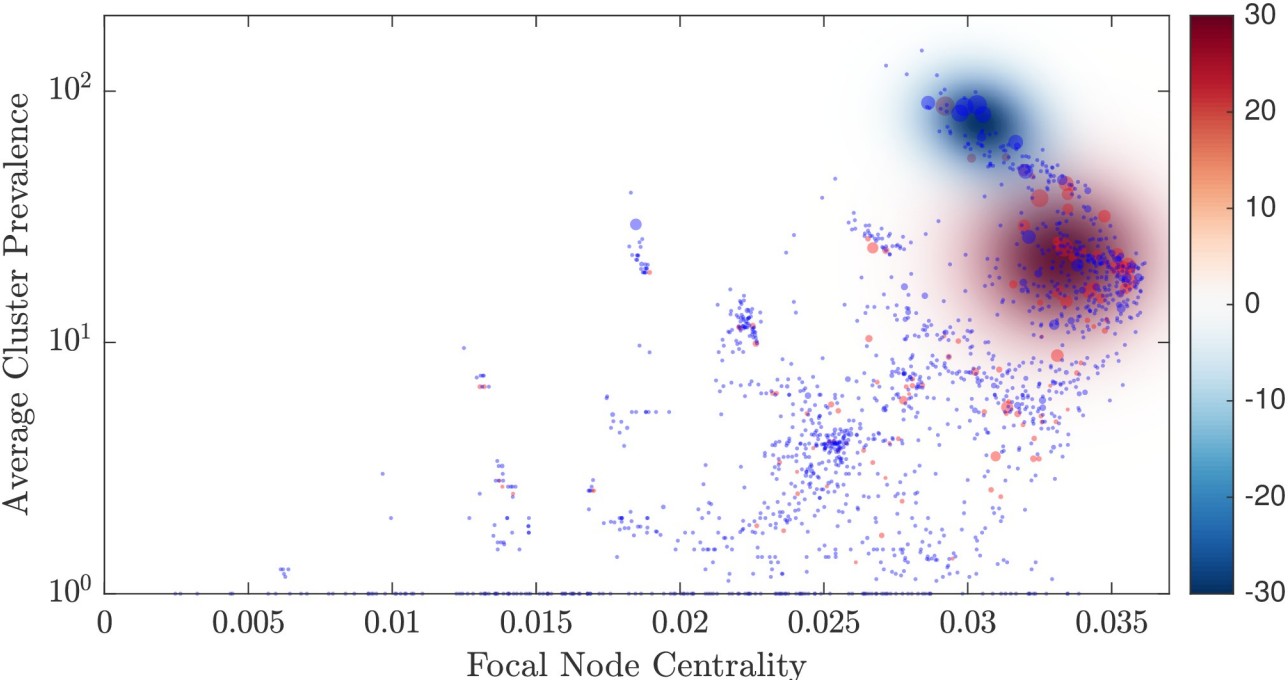

**Fig 5. The structure of the centrality-prevalence space, revealed by the expected values of both successful and unsuccessful evolutionary paths.** For every MLVA profile, the point size is shown in proportion to the average change in prevalence: positive change is shown with red colour, and negative change with blue. Intensity of each colour indicates the estimated probability density, given the average changes in prevalence for either positive or negative profiles. Similar to Fig 4, the transition region (intense red colour) is shaped by the successful profiles with high centrality and medium cluster prevalence. The evolutionary paths originating from profiles in this region tend to develop from right to left and from bottom to top, producing the higher average change in prevalence. Conversely, the bottleneck region (intense blue colour) is formed by unsuccessful profiles with the centrality around 0.03 and the cluster prevalence just below $10^2$. The evolutionary paths originating from these profiles tend to follow the opposite direction: from left to right and from top to bottom, reducing their prevalence on average.

also appear to concentrate within a particular region of the space, with relatively lower centrality around 0.025 and lower cluster prevalence below 10.

We also visualise the expected values of prevalence change along both successful and unsuccessful evolutionary paths separately: Fig 5 overlays these two densities, and S2 and S4 Figs show them independently, with S3 and S5 Figs illustrating the accuracy of the kernel density estimations. In addition to the transition region, which again is clearly identifiable, we can observe another dense region, formed by unsuccessful profiles with the centrality around 0.03 and the cluster prevalence just below 100. This region presents an emergent "barrier" for successful exploitative paths (RL-BT), as the profiles contained within it tend to develop in the opposite direction (LR-TB). In other words, not all successful exploitative paths proceed through this *bottleneck* region on the way to become the most virulent, while genetically more rare, strains.

Thus, the general evolutionary trend is reflected in the growing fitness of adapting pathogens (function) and the topology of the genotype networks (structure). While increasing the average cluster prevalence, this pathway initially follows from low to high centrality (left-to-right), and then reverses its direction from high to medium centrality (right-to-left). This function-structure correlation is also confirmed by Fig 6 which: (i) traces the average prevalence as a function of the distance to the most prevalent (reference) node, as well as (ii) compares the average prevalence with the size of connected components in the directed network. Firstly, the average prevalence is shown to decrease as the distance to the reference node grows, with a

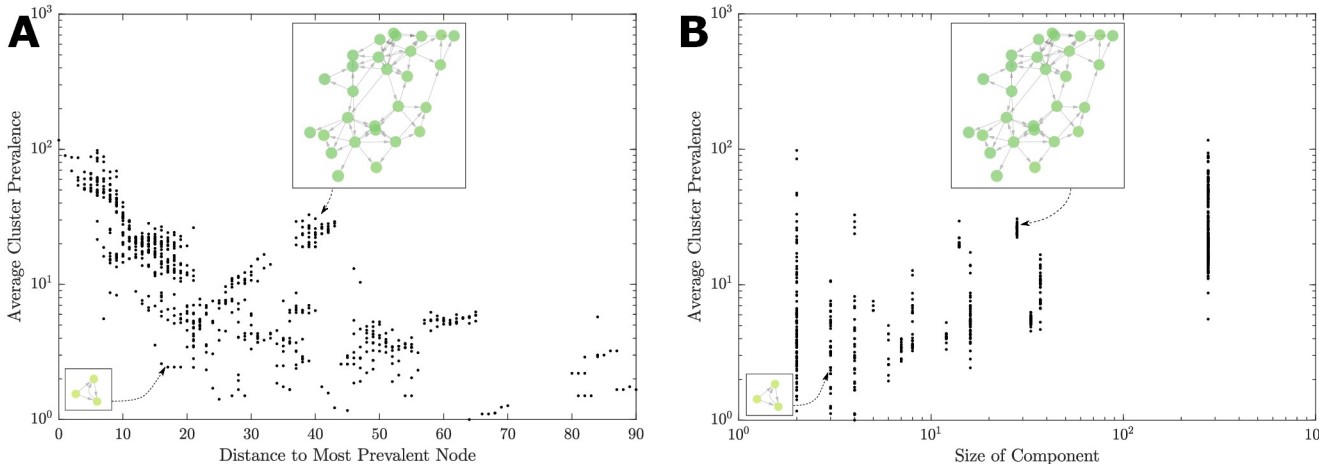

**Fig 6. Prevalence as a function of the genotype network characteristics.** Each point represents a node (MLVA profile), with the y-axis as the average prevalence of that profile's undirected cluster. A: this prevalence as a function of the distance to the reference node (the orange node in Figs 2 and 3). B: each node's prevalence is compared with the size of its connected-component in the directed network. The trend is punctuated: the prevalence increases in a step-wise fashion with decreasing distance (from successful strains), or increasing connected-component size (connectivity of the node). For the sample size of $N = 690$ connected nodes, the correlation between log prevalence and distance is $r = -0.613$ ($N = 690$, $p < 0.0001$) and between log prevalence and log component size is $r = 0.739$ ($N = 690$, $p < 0.0001$).

high correlation between log prevalence and distance, calculated as $r = -0.613$ ($N = 690$, $p < 0.00001$). In other words, the main evolutionary pathway is well-formed along both genetic branches, despite being somewhat punctuated at the low-to-medium centrality branch, as indicated by a step-wise character of the prevalence increases (Fig 6A). Secondly, as the prevalence grows, the average size of the connected components within the directed genotype network grows as well (Fig 6B). This high correlation between log prevalence and log component size (measured as $r = 0.739$ ($N = 690$, $p < 0.00001$)) indicates that the network itself becomes more interconnected along the main evolutionary pathway. It is interesting to contrast these two log-linear relationships with the nonlinear pathway-reversal in the prevalence-centrality space (Fig 3), where the centrality quantifies a more subtle topological role of individual nodes, rather than the more coarse connectedness within multi-node network components.

Quantifying the proportions of different general directions that the evolutionary pathways may take in the prevalence-centrality space has been equally illuminating for understanding evolutionary drives. There are 1,020 directed edges in the entire genotype network, and these edges yield 6,897 potential evolutionary paths. The absolute majority of the identified paths (66% of the paths, or 4,532 out of 6,897 paths) lead from right-to-left (reducing centrality) while going from bottom-to-top (increasing average prevalence). Furthermore, 72.4% of all paths originate from the transition region described above, and 64% of the paths starting in this region follow the RL-BT direction, amounting to 46.3% of all paths, see S6 Fig. We point out that there are only 2.87% of the all 1,673 nodes in the transition region. These observations strongly suggest a preferential evolutionary drive within the STM population (Fig 7). In other words, these paths evolve in a certain direction, RL-BT, in the centrality-prevalence space shown in Fig 3. We reiterate that the RL-BT direction is aligned with the "exploitative" branch pointed towards the most prevalent nodes. Furthermore, there is a significant correlation between the path length and the change in the average prevalence, measured as $r = 0.497$ ($N = 6,897$, $p = 0$), see S7 Fig. Importantly, on this evolutionary pathway, longer paths are more likely to achieve higher prevalence: as the path length increases, a higher number of paths are associated with this dominant direction (RL-BT). As Fig 7 illustrates, this is not the

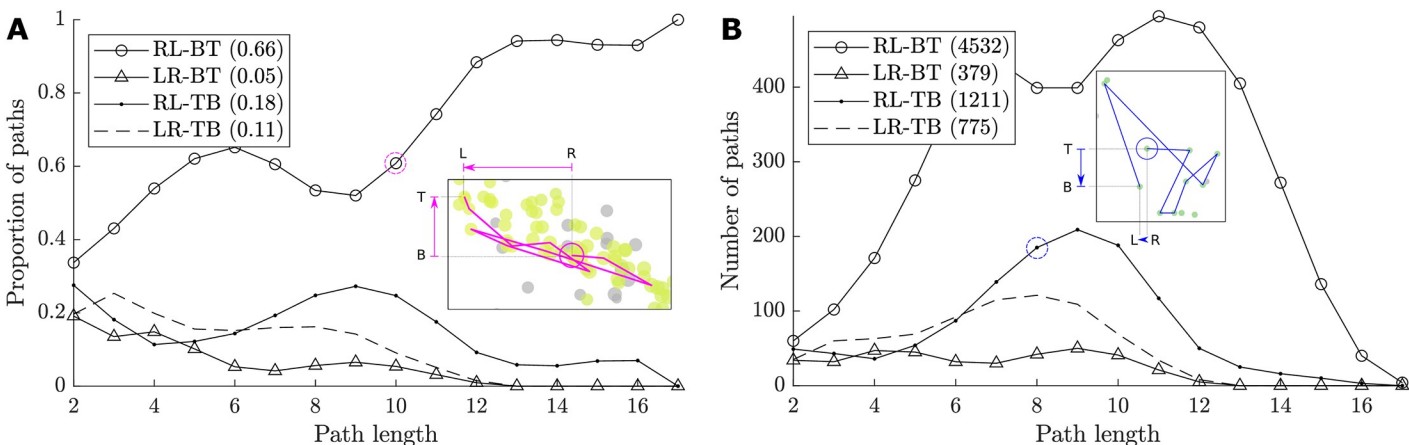

**Fig 7. Evidence for an evolutionary drive within the STM population.** The 1,020 edges in the genotype network yield 6,897 potential evolutionary paths. The subfigures show the proportion and number of paths evolving in a certain direction in the centrality-prevalence state space from Fig 3. This direction depends on the start and end-points of the path: right-to-left (RL), left-to-right (LR), top-to-bottom (TB), and bottom-to-top (BT). For instance, the first case study (magenta) is categorised as RL-BT (with ten edges), while the second case study (blue) follows the RL-TB direction (with eight edges). The absolute majority of paths (66%, or 4,532 paths) are shown to travel from the right-to-left (RL, decrease in centrality) and the bottom-to-top (BT, increase in prevalence), as shown in the figure legends. Further, the correlation between the path length and the change in prevalence is $r = 0.497$ ($M = 6,897$, $p < 0.00001$). A: as the path length increases, a higher number of paths are associated with this dominant direction. B: a majority of these paths are within the range of 6–12 edges; the smaller set of longer paths (above 12 edges) all decrease in centrality, mostly capturing exploitative adaptation (RL-BT) with a minority attributed to unsuccessful exploration (RL-TB).

case for three other possible directions, including the "explorative" branch pointed towards both higher prevalence and higher centrality (LR-BT).

To complement this analysis, Fig 7B shows that a majority of the paths are within the range of 6–12 edges. On the one hand, the smaller set of longer paths (above 12 edges) all decrease in centrality—this mostly corresponds to exploitative adaptation (RL-BT). On the other hand, another small set of shorter paths can be attributed to various directions of unsuccessful exploration. Thus, explorative paths tend to be shorter than exploitative ones.

We note that three considered case studies, associated with outbreaks previously investigated by the Australian Department of Health (see Materials and methods), include five, nine, and eleven STM strains, as shown in Figs 2 and 3. For example, the path shown in magenta passes through MLVA profile 3-10-7-14-523, which caused a significant outbreak in October 2013. This successful path is exploitative (RL-BT), with profile 3-10-7-14-523 appearing as the 9th node, within the bottleneck region. Another exploitative path, shown in red, starts at the profile 3-16-9-11-523, which is associated with at least two outbreaks in February 2014 and September 2015. The placement of this path on the upper branch indicates a strong potential for further outbreaks linked to its geneteic relatives along the path. We believe that the analysis of evolutionary paths in the centrality-prevalence space presents an effective means for classifying specific case studies and informing disease surveillance in general.

We point out that, without clustering, the centrality-prevalence space is less structured, as the individual profile prevalence is too noisy. We investigated several clustering thresholds $G_{max}$, and the structure of the centrality-prevalence space is preserved for clustering threshold $G_{max} \geq 3$, as shown in S11–S14 Figs, with the negative correlation between the distance to the most prevalent node and the log-prevalence growing in absolute terms with the clustering threshold $G_{max}$, from $r = -0.143$ ($N = 690$, $p = 1.62 \cdot 10^{-4}$) for $G_{max} = 0$ (singleton-clusters) to $r = -0.795$ ($N = 690$, $p < 0.00001$) for $G_{max} = 10$.

We compared the evolutionary paths and the preferential pathways for two different definitions of genetic distance. As detailed in Materials and methods, the multiple-loci distance, $G^{\bowtie}$,

captures *dispersed but limited* genetic variations across several loci, while the single-locus distance, $G^{\dagger}$, quantifies *local but variable* genetic variations only within a single locus. Thus, Fig 7, produced by using multiple-loci distance $G^{\bowtie}$, can be contrasted with S8 Fig, which presents corresponding results for single-locus distance $G^{\dagger}$. In the latter case, the correlation of path length to change in prevalence is not tangible at $r = -0.121$. Although the preferential pathway is still discernible in the direction (RL-BT), it is less dominant, with only 41% of the identified paths following it (Fig 7A). A majority of the paths which are identified with the single-locus distance $G^{\dagger}$ are within a range of shorter paths (4–8 edges), as shown by Fig 7B. This comparison suggests that the multiple-loci distance, $G^{\bowtie}$, which traces small genetic variations dispersed across several loci, is more suitable for identifying evolutionary paths.

We also showed that the preferential evolutionary pathway is independent of the role played by STTR3 locus in MLVA analysis. To do so, we again used two distinct definitions of genetic distance, $G^{\bowtie}$ and $G^{\dagger}$, applied to (i) the original pattern of MLVA profiles, with five loci, where STTR3 locus is left unaltered, and (ii) the adjusted pattern of MLVA profiles, with five loci, where STTR3 locus is corrected according to Larsson *et al.* [41], as explained in Materials and methods. The re-assigned MLVA profiles do not seem to add much more power to either the dominance of the preferential evolutionary pathway (RL-BT), or to the the correlation of path length to change in prevalence. For multiple-loci distance $G^{\bowtie}$, the correlation is larger and positive, $r = 0.269$ (see S9 Fig), and for single-locus distance $G^{\dagger}$, the correlation is small and negative, $r = -0.0474$ (see S10 Fig), but both values are less than the correlation of $r = 0.497$ (Fig 7A).

Interestingly, however, the multiple-locus distance $G^{\bowtie}$ applied to adjusted patterns with five loci, yields even stronger dominance of the preferential evolutionary pathway, with 76% of the paths following (RL-BT) direction. In addition, in this case, a clear majority of all identified paths sits in the range of longer paths with 8–16 edges (see S10 Fig). This indicates that the Larsson's correction of the fifth locus may add some power to analysis of evolutionary paths, but requires a more refined distance measure, in order to increase the correlation between the path length and the change in the average prevalence.

In summary, the comparative analysis across two distance measures and two representations of MLVA profiles showed that the preferential evolutionary pathway is dominant in all four combinations, with the best correlation given by the multiple-loci distance $G^{\bowtie}$ operating on the original MLVA pattern. For the Larsson's correction of the fifth locus, the multiple-loci distance $G^{\bowtie}$ also outperformed the single-locus distance $G^{\dagger}$. Furthermore, the dominance of the identified pathway was supported by observing a strong majority of longer paths contributing to evolutionary paths.

## Discussion

We described a model inferring genotype networks formed by co-evolving individual bacterial strains. This model explains dynamics of proliferating *Salmonella* strains, identifying their dominant evolutionary pathways, and highlighting the epidemiological importance of considering the population dynamics holistically. The results demonstrate how network-based analysis, facilitated through large-scale public health surveillance, can reveal salient evolutionary dynamics of infectious diseases.

The proposed approach, explicitly connecting the function and genetic structure of evolving pathogens, provides a radically new method for public health surveillance dealing with genetically diverse and spatiotemporally distributed epidemic scenarios. In contrast to existing mechanistic approaches based on the search for pathogens with matching genotypes, it enables

high-resolution monitoring of the population diversity of ongoing pathogens, identifying new genotypes as reservoirs from which future epidemics might emerge.

This method can be naturally extended to whole-genome sequencing (WGS) data, further increasing the resolution of evolutionary dynamics [43], as well as more general phylogenetic studies [44, 45]. Another direction of future research is to consider the modularity of inferred genetic networks, clarifying patterns of the punctuated evolution. Finally, emergence of dangerous pathogens may be compared with tipping points in complex systems, not only in terms of critical thresholds, but also in terms of the dynamic trajectories accumulating small changes over time [46, 47]. A comprehensive treatment of the evolutionary function-structure dynamics can explain how tipping points arise in the evolutionary dynamics of infectious diseases, quantify disease outbreaks with respect to critical thresholds, and determine precursors to tipping points.

## Materials and methods

### Multiple-locus variable-number tandem-repeats analysis (MLVA)

**MLVA**, as a method of genetic fingerprinting has proven sufficient to discriminate between STM strains for the purpose of epidemiological profiling by public health laboratory [31]. The multiplex Polymerase Chain Reaction (PCR) was performed to amplify the variable number tandem repeats (STTR9, STTR5, STTR6, STTR10 and STTR3). The method is, however, complicated by the fifth locus (known as STTR3), typically written instead as the entire locus fragment size because it includes both 27 bp and 33 bp repeat units.

We address this by considering, in addition, the MLVA re-assigning method outlined by Larsson *et al.* [41]. The reported STM genotypes are analyzed in two ways: (i) using the unchanged STTR3 locus, e.g., 3-9-7-12-523, as well as (ii) considering instead the sum of the corresponding 27 bp and 33 bp repeat units separately, e.g., 3-9-7-12-14, where 14 results from adding up the corresponding counts of tandem repeats 2 and 12. In the absence of a corresponding count of tandem repeats reported by Larsson *et al.* [41], we use a simple heuristic of zero 27 bp repeats, and 33 bp repeats equal to the rounded locus fragment size divided by 37, which serves as a close approximation to the reported values [41], e.g., 3-9-7-12-14.

**Case studies.**   Our analysis is illustrated by three case studies, focussed on MLVA profiles associated with some of the outbreaks previously investigated by the Australian Department of Health. The MLVA profile 3-10-7-14-523 was first detected in January 2010, and later caused a significant outbreak in October 2013 [48]. The MLVA profile 3-12-9-10-550 was first isolated in February 2008, and was later found to be the source of an outbreak in January 2011 [49]. The MLVA profile 3-16-9-11-523 first appeared in March 2010, and is associated with at least two outbreaks in February 2014 [50] and September 2015 [51].

### Network-based modelling

**Edge weights.**   The MLVA profile $i$ is given by the vector $p_i = \{p_{i1}, p_{i2}, \ldots, p_{iL}\}$, where the subscript denotes the loci, e.g., $p_{i1}$ is the number of repeats in locus 1 for MLVA profile $i$, and $L$ is 5 loci. Let $p_i$ and $p_j$ be the the MLVA profiles for nodes $i$ and $j$. Then, the $L_1$ norm distance $d_{ij}$, or the Manhattan distance, between these nodes can be computed as

$$d_{ij} = |p_i - p_j|_1 = |p_{i_1} - p_{j_1}| + |p_{i_2} - p_{j_2}| + \ldots + |p_{i_L} - p_{j_L}|.$$

We used the Manhattan distance in inferring the undirected network, analogously to our previous work [25]. Unlike [25] which used only the MLVA representation with Larsson's correction, here we consider both variants: the original representation and Larsson's correction. In

inferring the directed network, however, we define the distance $G$ between two MLVA profiles in two, more constrained, ways which capture two opposing perspectives. The first definition sets *multiple-loci distance*, $G_{ij}^{\bowtie}$, as the number of loci which differ by at most one tandem-repeat, that is, $1 \leq G^{\bowtie} \leq 5$. Consequently, if two MLVA profiles $i$ and $j$ differ in any locus by more than one tandem repeat, they are considered as unrelated (distance is $G_{ij}^{\bowtie}$ set to infinity).

The second definition, *single-locus distance*, $G_{ij}^{\dagger}$, restricts the differences between the tandem-repeats to only one locus, but allows profiles to disagree by more than one tandem-repeat. In other words, the multiple-loci distance captures *dispersed but limited* genetic variations across several loci, and may consider two profiles as genotypical neighbors if they differ by one tandem-repeat in each of their loci (for example, distance $G_{ij}^{\bowtie} = 3$ indicates that the two profiles differ in three loci, but only by one tandem-repeat in each). On other hand, the single-locus distance quantifies *local but variable* genetic variations, and would consider genetic proximity only when the profiles differ in one locus (for example, distance $G_{ij}^{\dagger} = 3$ means that the two profiles disagree only in one locus, but by three tandem-repeats).

For either of the distance definitions $G_{ij}$, without loss of generality, the edge weights are given by the reciprocal of these distances, i.e.,

$$w_{ij} = \frac{1}{G_{ij}}.$$

**Clustering.**   Using the edge weights, we follow the study [25] in obtaining the overlapping clustering of the undirected network nodes, by including all nodes within a certain distance from a given node. Formally, the cluster $C_i$ associated with node $i$ is given by $C_i = \{j: G_{ij} \leq G_{max}\}$, for some threshold $G_{max}$. Following [25], we set $G_{max} = 5$. Each node (i.e., profile) is associated with at least one cluster, even if that cluster only contains a single node. An interesting example is given by MLVA profile 3-9-7-12-622 (or 3-9-7-12-17 after the adjustment to match the study of Larsson *et al.* [41]): it has the individual prevalence of only 1, having been detected only once over the study period, but its average cluster prevalence is among the highest [25]. The reason is that there are multiple other MLVA profiles within the distance $G_{max} = 5$ from this profile, and some of these cluster neighbours show very high individual prevalence. The overlapping clustering algorithm that we employ aims to approximate the average virulence across such genetic neghbourhoods, which may correspond to various clonal complexes, groups, etc. S11–S14 Figs compare the centrality-prevalence spaces and the corresponding correlations, obtained for different thresholds $G_{max}$: 0, 3, 5, 10.

**Centrality.**   Then the closeness centrality of an individual node can be computed as the sum of the length of the shortest paths between nodes. In this work, as in [25], we use the normalized form of closeness centrality, across $N$ nodes:

$$l_i = \frac{N}{\sum_j G_{ij}}.$$

Within the network, we consider all directed paths, as unique sequences of directional steps. These paths are then classified as having one out of four possible directions, by considering relative centrality and relative cluster prevalence of their start and end points: Top vs Bottom, and Left vs Right, in the centrality-prevalence space. For example, a path with decreasing centrality (from Right to Left), and increasing prevalence (from Bottom to Top) would be classified as RL-BT.

## Bayesian inference of directed network

Assuming that the evolutionary/adaptive steps among the strains are reflected within the observed dataset, we built a statistical model to infer a directed genetic graph from the data. The graph $\mathcal{G} = \{\mathcal{V}, \mathcal{E}\}$ comprises a vertex set and an edge set. The set $\mathcal{V} = \{V_1, V_2, \ldots\}$ represents all STM strains in the dataset, with each vertex $V_i$ having an associated MLVA profile $P_i \in \mathbb{R}^5$ as a vector of five numbers as well as a sequence of instances $I_i = (I_{i1}, I_{i2}, \ldots)$, with $I_{in} \in \mathbb{Z}^+$ capturing the $n$th date that the profile $P_i$ was isolated. Our goal is to build the edge set $\mathcal{E} = \{E_1, E_2, \ldots\}$, where an edge $E_k = V_i \to V_j$ denotes a directed edge between two nodes $V_i$ and $V_j$.

To describe the inference of an edge, consider two vertices $V_p$ and $V_c$ of a graph that represent arbitrary MLVA profiles. We wish to infer the probability that the node $V_p$ is a potential parent (direct predecessor) of node $V_c$, i.e., *is there an edge $V_p \to V_c$?* The temporal proximity of $V_p$ to $V_c$ is denoted $\Delta$ and captures all detection dates of $V_p$ relative to the first detection of $V_c$ found in the dataset, i.e., $\Delta = (\Delta_1, \Delta_2, \ldots)$ where each $\Delta_n = I_{c1} - I_{pn}$. With this information, we aim to infer the likelihood of an evolutionary/adaptive step, e.g., a mutation, $M$, which represents the true-false outcome of whether $V_c$ evolved from $V_p$, for a given genetic distance $G$ (see Materials and methods for definitions of $G$). If the likelihood of this outcome is greater than a certain threshold $\Pr(M \mid \Delta, G) > \alpha$, then we include the edge $V_p \to V_c$ in the edge set $\mathcal{E}$.

Although we can not compute the distribution $\Pr(M \mid \Delta, G)$ directly from data, we can use Bayesian inference to infer it indirectly. A large majority of the dataset comprises non-mutations, i.e., self-sustaining outbreaks and continually circulating STM strains. Thus, instead of inferring the probability of $M$, we can compute the complementary event $\overline{M}$, i.e., the probability that $V_p$ is not a parent of $V_c$:

$$\Pr\left(\overline{M} \mid \Delta, G\right) = 1 - \Pr\left(M \mid \Delta, G\right). \tag{1}$$

Using Bayes formula, this can be reformulated into distributions that are easily obtained from data:

$$\Pr\left(\overline{M} \mid \Delta, G\right) = \frac{\Pr\left(\Delta \mid \overline{M}, G\right) \Pr\left(\overline{M} \mid G\right)}{\Pr\left(\Delta \mid G\right)}. \tag{2}$$

The main two distributions we are concerned with are $\Pr\left(\Delta \mid \overline{M}, G\right)$ and $\Pr(\Delta \mid G)$, since $\Pr\left(\overline{M} \mid G\right)$ is a constant over each genetic distance. That is, we can compute the distribution as:

$$\Pr\left(\overline{M} \mid \Delta, G\right) \propto \frac{\Pr\left(\Delta \mid \overline{M}, G\right)}{\Pr\left(\Delta \mid G\right)}. \tag{3}$$

Now, $\Pr(\Delta \mid G)$ can simply be computed by choosing every genetic distance in the dataset and looking at all relative occurrences. Using these data, we build a frequency distribution of all relative occurrences $\Delta$ within each genetic distance $G$; normalizing then gives us the probability mass function $\Pr(\Delta \mid G)$. In contrast, $\Pr\left(\Delta \mid \overline{M}, G\right)$ can be computed from scenarios where it is highly unlikely that there was a mutation. These highly-unlikely scenarios are all instances *after* the first detection of an STM strain. Computing these two functions, taking their ratio and normalizing then gives the probability $\Pr(M \mid \Delta, G)$ of there being a mutation within a certain number of days $\Delta$ for each genetic distance $G$. Fig 8 illustrates the statistical inference algorithm.

As stated, we now include the edge $V_p \to V_c$ if $\Pr(M \mid \Delta, G) > \alpha$, where $\alpha$ is some statistical significance threshold. However, each child (direct successor) could pertain to a number of

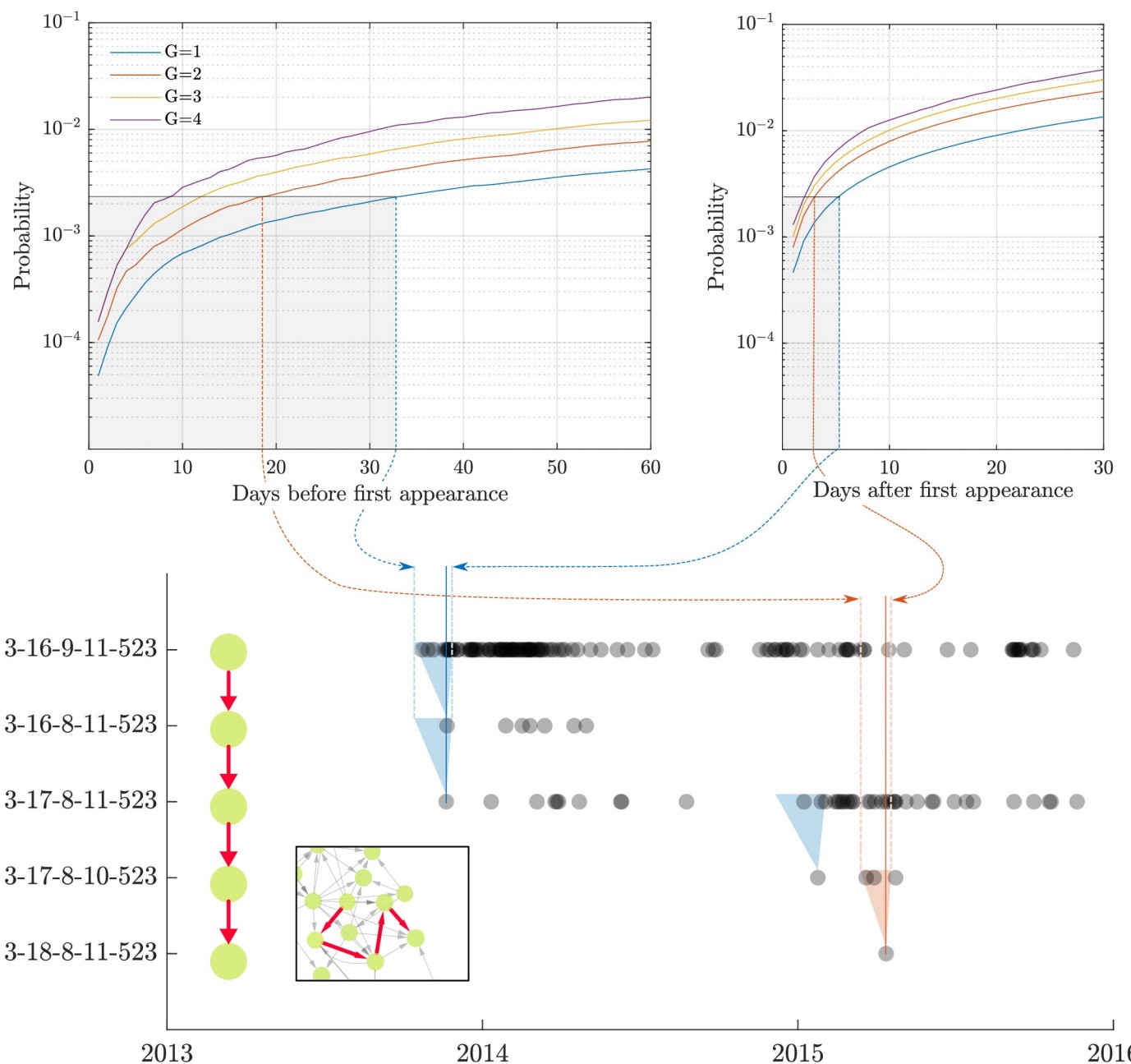

**Fig 8. Inference of an evolutionary step used in constructing edges in the directed genotype network.** For every genetic distance $G$, we use Bayesian inference to determine two temporal intervals that indicate potential evolutionary precedence. These intervals capture parent nodes that were isolated either before a child node (pre-windows) or after a child node (post-windows). The arcs in the top subfigures are cumulative probability distributions (CDFs) of these intervals for different genetic distances (number of loci modified) $G$. For a given level of (multiple comparisons corrected) statistical significance and genetic distance, the suitable pre- and post-windows are obtained by inverting the CDF; shown as dashed blue and red lines. Given first detection of a potential offspring MLVA profile, all nodes which are present during these intervals generate a directed edge to the offspring. The bottom subfigure illustrates inference of the four edges of the third case study, where the first three evolutionary steps crossed genetic distances of $G = 1$, with the final step crossing two loci ($G = 2$). Shaded blue and red triangles show the corresponding temporal intervals (i.e., combined pre- and post-windows) for $G = 1$ (blue) and $G = 2$ (red), in relation to the observational data points marked by vertical solid lines.

potential parents for a given distance $G$, and so we are performing multiple hypothesis tests at once. Thus, the significance threshold must be adjusted to account for the higher number of false positives expected under this scenario. For this work, we use Bonferroni correction as a conservative correction, which results in the threshold $\alpha$ being divided by the number of potential parents we are testing for each child at every genetic distance $G$.

## Supporting information

**S1 Fig. The cumulative density function (CDF) of the expected value of the positive gain in cluster prevalence (using the kernel density estimation): Actual vs estimated.** The contributions of all unsuccessful paths are bounded to 1.
(TIF)

**S2 Fig. Prevalence change for successful evolutionary paths.** The structure of the centrality-prevalence space, revealed by the expected value of only successful evolutionary paths. For every MLVA profile, the point size is shown in proportion to the average (positive) change in prevalence. Colour intensity indicates the estimated density, given the average positive changes in prevalence. Similar to Fig 5, the transition region (intense red colour) is shaped by the successful profiles with high centrality and medium cluster prevalence. The evolutionary paths originating from profiles in this region tend to develop from right to left and from bottom to top, producing the higher average change in prevalence.
(TIF)

**S3 Fig. The cumulative density function (CDF) of the expected value of the cluster prevalence change along successful evolutionary paths (using the kernel density estimation): Actual vs estimated.**
(TIF)

**S4 Fig. Prevalence change for unsuccessful evolutionary paths.** The structure of the centrality-prevalence space, revealed by the expected value of only unsuccessful evolutionary paths. For every MLVA profile, the point size is shown in proportion to the average (negative) change in prevalence. Colour intensity indicates the estimated density, given the average negative changes in prevalence. Similar to Fig 5, the bottleneck region (intense blue colour) is formed by unsuccessful profiles with the centrality around 0.03 and the cluster prevalence just below $10^2$. The evolutionary paths originating from these profiles tend to develop from left to right and from top to bottom, reducing their prevalence on average.
(TIF)

**S5 Fig. The cumulative density function (CDF) of the expected value of the cluster prevalence change along unsuccessful evolutionary paths (using the kernel density estimation): Actual vs estimated.**
(TIF)

**S6 Fig. Expected number of paths.** The structure of the centrality-prevalence space, revealed by the expected number of evolutionary paths. For every MLVA profile, the point size is shown in proportion to the number of paths that start from this profile. Colour intensity indicates the estimated density.
(TIF)

**S7 Fig. Length–prevalence correlation.** Correlation between the path length and the change in the average prevalence, measured as $r = 0.497$ ($N = 6, 897$, $p < 0.00001$).
(TIF)

**S8 Fig. Paths: Unadjusted (one locus).** Constrained to genetic variations within one locus only. Number of nodes $N = 718$, number of edges is 967, and the number of paths $M = 2,304$. Correlation of path length to change in prevalence: $r = -0.121$ ($M = 2,304$, $p < 0.00001$). (TIF)

**S9 Fig. Paths: Larsson's correction (multiple loci).** Constrained to genetic variations over multiple loci with Larsson's correction. Number of nodes $N = 744$, number of edges is 1,190, and the number of paths $M = 23,915$. Correlation of path length to change in prevalence: $r = 0.269$ ($M = 23,915$, $p < 0.00001$). (TIF)

**S10 Fig. Paths: Larsson's correction (single locus).** Constrained to genetic variations within one locus only with Larsson's correction. Number of nodes $N = 730$, number of edges is 991, and the number of paths $M = 2,323$. Correlation of path length to change in prevalence: $r = -0.0474$ ($M = 2,323$, $p = 0.0224$). (TIF)

**S11 Fig. Clustering threshold 0.** The centrality-prevalence space for clustering threshold $G_{max} = 0$ (singleton-clusters). The centrality is measured with respect to the undirected genotype network. The colour shows the genetic distance of a profile to the reference profile (the most prevalent node), with grey points indicating disconnected profiles in the directed network. Correlation between the distance to the most prevalent node and the log-prevalence $r = -0.143$ ($N = 690$, $p = 1.62 \cdot 10^{-4}$). Correlation between the log-component-size and log-prevalence: $r = 0.332$ ($N = 690$, $p < 0.00001$). (TIF)

**S12 Fig. Clustering threshold 3.** The centrality-prevalence space for clustering threshold $G_{max} = 3$. The centrality is measured with respect to the undirected genotype network. The colour shows the genetic distance of a profile to the reference profile (the most prevalent node), with grey points indicating disconnected profiles in the directed network. Correlation between the distance to the most prevalent node and the log-prevalence $r = -0.513$ ($N = 690$, $p < 0.00001$). Correlation between the log-component-size and log-prevalence: $r = 0.734$ ($N = 690$, $p < 0.00001$). (TIF)

**S13 Fig. Clustering threshold 5.** The centrality-prevalence space for clustering threshold $G_{max} = 5$. The centrality is measured with respect to the undirected genotype network. The colour shows the genetic distance of a profile to the reference profile (the most prevalent node), with grey points indicating disconnected profiles in the directed network. Correlation between the distance to the most prevalent node and the log-prevalence $r = -0.613$ ($N = 690$, $p < 0.00001$). Correlation between the log-component-size and log-prevalence: $r = 0.739$ ($N = 690$, $p < 0.00001$). (TIF)

**S14 Fig. Clustering threshold 10.** The centrality-prevalence space for clustering threshold $G_{max} = 10$. The centrality is measured with respect to the undirected genotype network. The colour shows the genetic distance of a profile to the reference profile (the most prevalent node), with grey points indicating disconnected profiles in the directed network. Correlation between the distance to the most prevalent node and the log-prevalence $r = -0.795$ ($N = 690$, $p < 0.00001$). Correlation between the log-component-size and log-prevalence: $r = 0.683$ ($N = 690$, $p < 0.00001$). (TIF)

## Acknowledgments

We are thankful to Joseph Lizier for his advice on kernel density estimations.

## Author Contributions

**Conceptualization:** Oliver M. Cliff, Vitali Sintchenko, Tania C. Sorrell, Mikhail Prokopenko.

**Data curation:** Vitali Sintchenko.

**Formal analysis:** Oliver M. Cliff, Mikhail Prokopenko.

**Funding acquisition:** Vitali Sintchenko, Tania C. Sorrell, Mikhail Prokopenko.

**Investigation:** Oliver M. Cliff, Natalia McLean, Vitali Sintchenko, Mikhail Prokopenko.

**Methodology:** Oliver M. Cliff, Natalia McLean, Vitali Sintchenko, Stuart Kauffman, Mikhail Prokopenko.

**Project administration:** Mikhail Prokopenko.

**Resources:** Mikhail Prokopenko.

**Software:** Oliver M. Cliff, Natalia McLean, Kristopher M. Fair.

**Supervision:** Mikhail Prokopenko.

**Visualization:** Oliver M. Cliff, Mikhail Prokopenko.

**Writing – original draft:** Oliver M. Cliff, Mikhail Prokopenko.

**Writing – review & editing:** Oliver M. Cliff, Natalia McLean, Vitali Sintchenko, Kristopher M. Fair, Tania C. Sorrell, Stuart Kauffman, Mikhail Prokopenko.

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
