## [Decision Letter · Decision Letter 0]

5 Jul 2020

Dear Prof. Dr. Prokopenko,

Thank you very much for submitting your manuscript "Inferring evolutionary pathways and directed genotype networks of foodborne pathogens" for consideration at PLOS Computational Biology.

As with all papers reviewed by the journal, your manuscript was reviewed by members of the editorial board and by several independent reviewers. In light of the reviews (below this email), we would like to invite the resubmission of a significantly-revised version that takes into account the reviewers' comments. Both reviewers were favorable about your work, but made numerous substantive suggestions about how the content could be strengthened.

We cannot make any decision about publication until we have seen the revised manuscript and your response to the reviewers' comments. Your revised manuscript is also likely to be sent to reviewers for further evaluation.

Sincerely,

James Lloyd-Smith

Associate Editor

PLOS Computational Biology

Rob De Boer

Deputy Editor

PLOS Computational Biology

Reviewer's Responses to Questions

**Comments to the Authors:**

Reviewer #1: Cliff et al. developed a directed network-based approach to model the evolution of Salmonella Typhimurium (STM). A longitudinal dataset across a 10-year period in New South Wales was used. Each STM isolate was genotyped by multiple-locus variable-number tandem-repeats analysis, which was an established method. This dataset contained 17,107 STM isolates (99.3% of all isolated in that 10-year period), with 1,674 unique MLVA profiles. This dataset was collected previously and has been analyzed by the authors (reference #18 of the manuscript). A directed genotype network was established using both temporal and genetic proximity information between MLVA profiles. In addition, an undirected network was also built by clustering similar MLVA profiles following the methods described in a previous study published by the authors (reference #18 of the manuscript). The directed genotype network allowed evolutionary pathways to be traced, whereas the undirected network provided information on centrality and prevalence (the average prevalence of genetically close MLVA profiles). Increase in prevalence along an evolutionary pathway is interpreted as increase in fitness, whereas decrease in centrality is interpreted as exploring ecological niches. The authors found that Most directed edges reduce centrality while increase prevalence, indicating that exploitative adaptation is common. In addition, longer paths are more likely to achieve higher prevalence.

Overall, this study is an improved version of the published analysis by the authors (ref 18 in the manuscript), by using Bayesian inference to build the directed genotype network. The approach is quite nice and the results provide interesting insight about the evolution of STM. As described in the discussion, the approach developed in this study may be extended to analyze datasets other than MLVA profiles, such as genome sequencing data. However, there are several issues need to be addressed as stated below.

Specific comments:

1. Is the MLVA dataset described in this study publicly available?

2. Is the undirected network described in this study identical to the one described in the previous study (ref 18 in the manuscript)? Please clarify.

3. Can the author highlight the reference node in Figure 1? It will be interesting to know when the most prevalent genotype appears in time.

4. In line 223: It is unclear to me how the author defined “a typical evolutionary path”. Was it simply based on decreasing genetic distance from the reference node (color gradient)? It seems to me the claims in lines 223 to 231 about “a typical evolutionary path” were based on the analysis of undirected network without using the information from the directed network?

5. In lines 263-265: “Furthermore, there is a significant correlation between the path length and the change in the average prevalence, measured as r = 0.498 (N = 6,896, p < 0.00001).” Can the author provide a figure of this analysis for visualization?

6. In lines 267-269: “As Figure 4 illustrates, this is not the case for three other possible directions, including the “explorative” branch pointed towards both higher prevalence and higher centrality (LR-BT).” Can the author elaborate a bit more on this point? It is not clear to me how the author arrives to the conclusion that the “explorative” branch pointed towards LR-BT.

7. The three case studies were mentioned in the first paragraph of the result section, but were not described later on in the text. Is there anything being learned from the case studies?

8. Legend of Figure 3 may need to be organized a bit? There is a paragraph of figure legend under each panel of Figure 3. But there is also another paragraph of figure legend at the bottom of the page.

Reviewer #2: Review uploaded as attachment.

**Have all data underlying the figures and results presented in the manuscript been provided?**

Reviewer #1: **No: **The MLVA dataset is not provided.

Reviewer #2: **No: **The review package I got specifies the details.

PLOS authors have the option to publish the peer review history of their article (what does this mean?). If published, this will include your full peer review and any attached files.

Reviewer #1: No

Reviewer #2: No
---

## [Decision Letter · Decision Letter 1]

16 Sep 2020

Dear Prof. Dr. Prokopenko,

Thank you very much for submitting your manuscript "Inferring evolutionary pathways and directed genotype networks of foodborne pathogens" for consideration at PLOS Computational Biology. As with all papers reviewed by the journal, your manuscript was reviewed by members of the editorial board and by several independent reviewers. The reviewers appreciated the attention to an important topic. Based on the reviews, we are likely to accept this manuscript for publication, providing that you modify the manuscript according to the review recommendations.

In particular, reviewer 2 outlines two specific passages that they feel would benefit from minor text revisions.  If you are able to make these changes, and provide with a tracked changes format as they request, I am confident that this final round of review can be handled expeditiously. 

Sincerely,

James Lloyd-Smith

Associate Editor

PLOS Computational Biology

Rob De Boer

Deputy Editor

PLOS Computational Biology

[LINK]

Reviewer's Responses to Questions

**Comments to the Authors:**

Reviewer #1: The authors have addressed all my previous concerns.

Reviewer #2: The manuscript is strongly improved. However, a few un-refined passages are holding it back from the clarity needed for publication just yet. Once these below points are addressed, I would happily recommend publication.

1. The clarified description of the clustering is much better at communicating how the clustering works. However, I feel that this clarity is not properly reflected in Fig. 2. While I realize it can be difficult to scope caption descriptions to be sufficiently descriptive yet also concise, I believe that because Fig. 2 is first mentioned so early in the manuscript, a bit of additional clarity would be beneficial for readers. For example, lines 464-466 quickly and clearly convey the fact that there is formally a cluster calculated for each node, even if two nodes can have identical clusters. I believe that the somewhat ambiguous use of the word "associated" in the caption caused me confusion, because while in the formal definition it is clear that every node is assigned a cluster, in normal language I would also consider nodes that a cluster contains to be "associated" with that cluster (even though formally the cluster is representative of a single specific node). It is also confusing saying that "each profile may be associated with a cluster of other profiles". The methods make it clear that a cluster is assigned to every node, even if that cluster is an empty set. To make a minimal concrete suggestion: the sentence "Formally, the cluster Ci associated with node i is given by C i= { j : G ij≤ G max} ," should just be included in the caption. Perhaps it seems I am belaboring this point, but I think it's important, since this is the first time the reader is exposed to how the written research handles clustering, and it plays into so many of the analyses and conclusions.

2. I very much appreciate the extra descriptions and analyses the authors have provided to clarify what is termed the "transition region" (line 257). However, I feel that this new-found clarification is undermined by it being presented on top of the existing un-clarified discussion of the "two well-defined branches in the space formed by node centrality and cluster prevalence", rather than integrated with this definition, or replacing it entirely. One of my original comments was that these described regions were "very ambiguous as written" and I recommended "that since the "branches" as described play a key role in the results, they should be highlighted in any figures they are visible in." Unfortunately, the 3rd and 4th paragraphs of the results section appear completely unchanged (lines 224-248), and the referenced figure 3 still gives no visible indications or explicit descriptions of where these two "well-defined" branches are, or where the "plateaus" are. This might not be as big a problem if they weren't the first results that the reader is informed of. I believe that communication of the results would be greatly improved if lines 224-248 either forgo reference to the ill-defined branches, or if the branches are made explicit--including their relation to the "transition region"--using either visual cues/text on the figure, and/or additional explicit main-text description.

3. One last request--if the authors could kindly highlight their changes in the final revised submission, the review could be completed more expeditiously.

**Have all data underlying the figures and results presented in the manuscript been provided?**

Reviewer #1: None

Reviewer #2: Yes

PLOS authors have the option to publish the peer review history of their article (what does this mean?). If published, this will include your full peer review and any attached files.

Reviewer #1: No

Reviewer #2: No
---

## [Editor Report · Decision Letter 2]

25 Sep 2020

Dear Prof. Dr. Prokopenko,

We are pleased to inform you that your manuscript 'Inferring evolutionary pathways and directed genotype networks of foodborne pathogens' has been provisionally accepted for publication in PLOS Computational Biology.

Best regards,

James Lloyd-Smith

Associate Editor

PLOS Computational Biology

Rob De Boer

Deputy Editor

PLOS Computational Biology

---

## [Editor Report · Acceptance letter]

23 Oct 2020

PCOMPBIOL-D-20-00724R2 

Inferring evolutionary pathways and directed genotype networks of foodborne pathogens

Dear Dr Prokopenko,

I am pleased to inform you that your manuscript has been formally accepted for publication in PLOS Computational Biology. Your manuscript is now with our production department and you will be notified of the publication date in due course.

With kind regards,

Nicola Davies
